# SDePR: Fine-Grained Leaf Image Retrieval with Structural Deep Patch Representation

## ABSTRACT

Fine-grained leaf image retrieval (FGLIR) is a new unsupervised pattern recognition task in content-based image retrieval (CBIR). It aims to distinguish varieties/cultivars of leaf images within a certain plant species and is more challenging than general leaf image retrieval task due to the inherently subtle differences across different cultivars. In this study, we for the first time investigate the possible way to mine the spatial structure and contextual information from the activation of the convolutional layers of CNN networks for FGLIR. For achieving this goal, we design a novel geometrical structure, named Triplet Patch-Pairs Composite Structure (TPCS), consisting of three symmetric patch pairs segmented from the leaf images in different orientations. We extract CNN feature map for each patch in TPCS and measure the difference between the feature maps of the patch pair for constructing local deep self-similarity descriptor. By varying the size of the TPCS, we can yield multi-scale deep self-similarity descriptors. The final aggregated local deep self-similarity descriptors, named Structural Deep Patch Representation (SDePR), not only encode the spatial structure and contextual information of leaf images in deep feature domain, but also are invariant to the geometrical transformations. The extensive experiments of applying our SDEPR method to the public challenging FGLIR tasks show that our method outperforms the state-of-the-art handcrafted visual features and deep retrieval models.

## CCS CONCEPTS

• **Computing methodologies**;

## KEYWORDS

Fine-grained leaf image retrieval, object image description, deep convolutional feature, structural feature representation

## 1 INTRODUCTION

Conventional leaf image retrieval is the task of searching for leaf images that belong to the same species as the query leaf. It has attracted long-term attention [10, 11, 16, 27, 28] in the computer vision research community due to its significant role in biological research and biodiversity protection. In recent years, with the rapid development of crop cultivation technology, more and more crop varieties/cultivars have been cultivated in modern agriculture system.

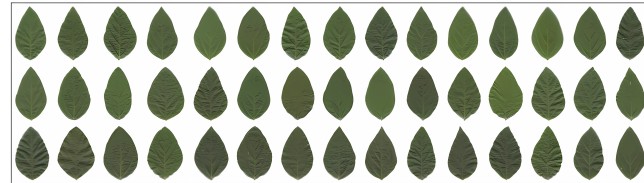

**Figure 1: An example fine-grained leaf image collection to show the high similarity of leaf image patterns across different cultivars (the leaf samples are from the different soybean cultivars).**

For instance, there have been about 2500 soybean cultivars released since the scientific breeding of soybean cultivars began in the early 20th century [18]. Fig. 1 presents some typical leaves of different soybean cultivars. It can be seen that there are very high similarity of leaf image patterns across different cultivars which accordingly have caused increasing concerns [12, 25, 29, 30] about whether leaf image pattern can be effectively used for distinguishing cultivars.

Unlike conventional leaf image retrieval task that focuses on distinguishing spe-cies, fine-grained leaf image retrieval (FGLIR) aims at retrieving leaf images belong-ing to the diverse cultivars of a certain plant species (e.g. soybean) and returning leaf images with the same cultivar as the query leaf image [5, 6]. Due to the great high inter-class similarity as shown in Fig. 1, FGLIR task requires cultivar-level leaf image similarity, i.e., fine-grained image similarity in which two images are considered similar if and only if they belong to the same cultivar. Mining discriminative leaf image features is at the core of finding fine-grained image similarity. A desirable leaf image feature representation is expected to narrow the intra-class distance while increase the inter-class distance as much as possible in image feature space.

In the past decades, a large body of leaf image descriptors have been devoted to plant species or cultivar recognition tasks. Most of them are based on handcrafted features which take shape, texture, vein, or edge patterns of leaf images as main clues. Although some methods [4, 21, 22] have reported high accuracies on species recog-nition, they are still limited to capture the subtle difference of leaf image pattern across different cultivars in FGLIR task due to the intrinsic essence of handcrafted features heavily depending on the human expertise. Inspired by the great success of deep learning in various image recognition tasks, some recent attempts [9, 15, 32] have been made to employ deep learning features to leaf image recognition. However, these methods focuses on leaf image classi-cation, a supervised pattern recognition task, and their performance depend on a large body of labeled training samples.

FGLIR is a typical unsupervised pattern recognition task. Its characteristics of fine-grained leaf images and unlabeled dataset make it more challenging than other leaf image recognition tasks. Although some recent efforts [14, 17, 19, 34] have been made to

employ deep features for image retrieval, few attention has been paid on leveraging deep features for FGLIR. In this paper, we investigate the possible way to enhance the discriminative power of deep convolutional features for setting a new state-of-the-art performance on fine-grained leaf image retrieval.

Image descriptors produced by deep convolutional neural networks (CNN) have emerged as state-of-the art generic descriptors for various visual recognition tasks [7, 23, 31]. Our work also relies on using CNN activations as off-the-shelf features for addressing the challenging FGLIR. Some CNN based image retrieval methods [20, 27, 38] focus on extracting feature from the fully-connected layers. However they are limited to capture the spatial structural features of images due to the loss of spatial information in the fully-connected layers. Recently, most of methods [2, 14, 19, 31, 34, 35], concentrate on building image descriptors from the convolutional layers. Some of them [2, 19, 31] mainly pay attention on the feature choice or aggregation of the convolutional layers and ignore the useful spatial structure feature extraction. Although some recent at-tempts [14, 34, 35] have been made to encode the spatial structure and contextual information from feature maps of the convolutional layers into the final image descriptors, they require additional labeled datasets to train the networks.

In this work, we propose a novel CNN feature based leaf image feature representa-tion, named Structural Deep Patch Representation (SDePR), for fine-grained leaf image retrieval (FGLIR). The contributions of this study is highlighted as follows: (1) We for the first time investigate the possible way to construct leaf image descriptors that can encode the spatial structure and contextual information from the feature maps of the convolutional layers in a training-free manner. (2) We design a triplet patch-pair composite structure (TPCS), to measure the local structure properties and spatial correlations of leaf images. (3) We conduct extensive experiments to apply our proposed SDePR method to the public challenging fine-grained leaf image da-tasets and greatly improve the retrieval rates of the state-of-the-art methods.

## 2 RELATED WORK

In this section, we briefly survey the state-of-the-art leaf image descriptors for FGLIR and deep CNN feature based image retrieval methods.

### 2.1 Leaf Image Descriptor

The early leaf image descriptors are designed for plant species recognition and mainly extract various shape features [10, 11, 16, 26–28] to construct feature representations. The leaf texture features are also considered in this task. However, they are only taken as the complementary clues for enhancing the recognition accuracy [1, 34]. While in FGLIR, as shown in Fig. 1, the leaf images have high intra-class similarity in their shape patterns. So, the existing methods for FGLIR mainly focuses on the use of no-shape features.

Larese et al. [13] used the Self-Invariant Feature Transform (SIFT) to detect vein patterns and employed Bag of Words (BoW) model to aggregate local vein features for distinguishing soybean cultivars. Oleander has many cultivars exhibiting high inter-class similarity. Baldi et al. [3] used 18 morphometric and colorimetric parameters as leaf visual features for distinguishing 22 oleander cultivars.

Recently, some methods treat leaf texture patterns as the most valuable clues for FGLIR. Wang et al. [29] proposed a novel local R-symmetry co-occurrence to de-picting discriminative local symmetry texture patterns. Chen et al. [5] designed a novel geometrical configuration, named Symmetric Binary Tree (SBT), for mining co-occurrence texture patterns. More recently, based on the fan-beam projection theory, a new texture descriptor, named Fan-Beam Binarization Difference Projec-tion (FB-BDP) [6], was developed for various FGLIR tasks. A single leaf image pat-tern may be not powerful enough to distinguish cultivars. Wang et al. [30] for the first time constructed leaf image descriptors from the joint leaf image patterns in which the leaves from the lower, middle and upper parts of plants are combined for joint texture feature extraction.

Besides the handcrafted methods, some works have also considered the use of deep learning for identifying cultivars. Tavakoli et al. [25] proposed to use convolu-tional neural networks (CNN) to classify 12 cultivars of common beans that belong to three different species. They fine-tuned the pre-trained VGG16 model [24] by replacing the last two dense layers with two new ones of 1024 and 512 neurons. However it is designed to work in the supervised setting, i.e., annotated images are required for model fine-tuning. While FGLIR is a pure unsupervised recognition task and annotated images are unavailable. Therefor this method cannot handle the challenging FGLIR.

### 2.2 CNN based Image Retrieval

According to whether the deep feature representations encode the structural in-formation of images, the existing CNN based image retrieval methods can be classified into non-structural method and structural method. Most of the methods to the former. They generally focuses on using the activations from the fully connected layers or convolutional layers of CNN models and adopting various feature aggregat-ing strategies to build image descriptors. Wei et al. [31] proposed a method, named Selective Convolutional Descriptor Aggregation (SCDA), for fine-grained image retrieval. It takes the activations from the convolutional layers of CNN models as clues to select the useful deep descriptors for feature aggregation. Pang et al. [19] proposed to address deep convolutional feature aggregation by simulating the dynamics of heat diffusion. Yang et al. [34] made the first attempt to fuse local and global convolutional layer features in an orthogonal manner for effective single-stage image re-trieval. They proposed a novel information fusion framework, named Deep Orthogonal Local and Global (DOLG), for achieving this goal.

Capturing the structural characteristics of image can enhance the discriminative power and robustness of image descriptors [8, 14]. Instead simply computing the CNN activation vector over the entire image, Gong et al. [8] proposed to combine activations of CNN extracted at multiple local image windows for getting improved recognition performance. Self-similarity is a classical structural image descriptor aim-ing to measure how similar a specific part of an image is to the entire image or its neighborhood region [14]. Lee et al. [14] revisited it in terms of convolutional and proposed a self-similarity encoder that embeds structural properties in global embeddings while learning diverse structural properties within numerous images.

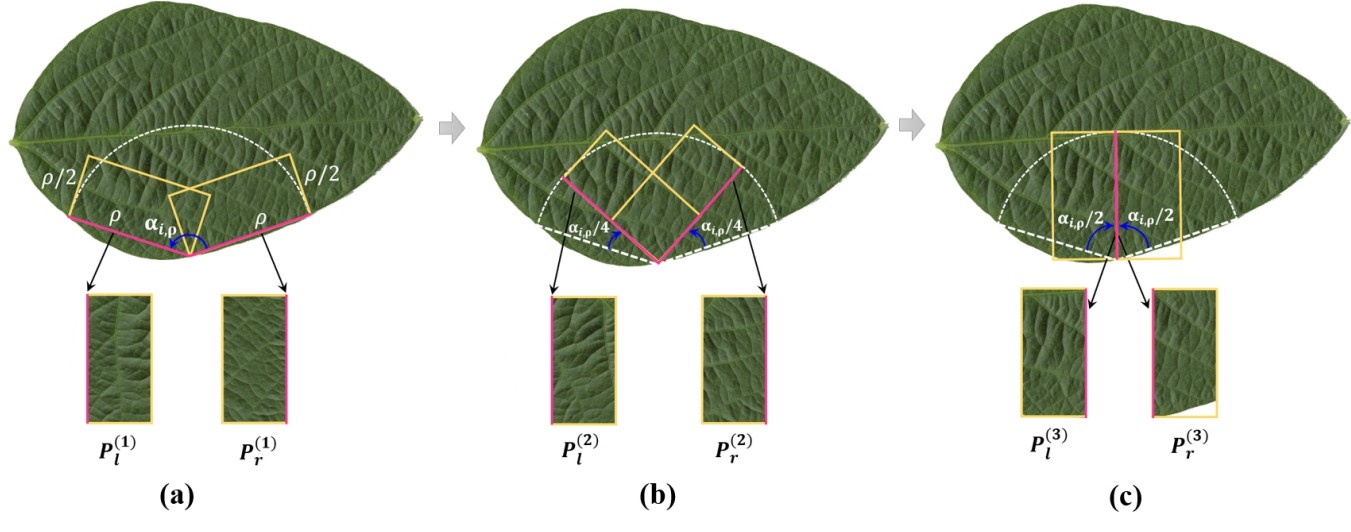

**Figure 2: An Illustration of constructing the triplet patch-pairs composite Structure (TPCS).**

## 3 THE PROPOSED METHOD

In this section, we introduce the details of our proposed Structural Deep Patch Representation (SDePR) for fine-grained leaf image retrieval (FGLIR).

### 3.1 Triplet Patch-Pairs Composite Structure (TPCS)

Given a leaf image $I$, let $\Omega$ denote the leaf contour extracted from it. We uniformly sample $N$ points, $Z_i, i = 0, \ldots N - 1$, from the leaf contour $\Omega$. Then the original leaf contour $\Omega$ is simplified as a polygon with $N$ edges. For each sample point $Z_i$, the largest distance between it and the other $N - 1$ sample ones is defined as

$$\Upsilon_i = \max_{j \in [0, N-1]} \|z_j - z_i\|_2, \qquad (1)$$

where $\| \cdot \|_2$ denotes L2-norm. Given a length $\rho \in (0, R_i)$, we use it as a radius and the contour point $Z_i$ as the center to draw a circle. Since $\rho < \Upsilon_i$, the circle intersects with the contour at least two points. Among all the intersection points, let $Z_l$ and $Z_r$ be separately the nearest neighboring points of the point $Z_i$ along contour in clockwise and counter-clockwise directions. We denote $\alpha_{i,\rho}$ as the angle formed by rotating the right chord $Z_i Z_r$ about the point $Z_i$ to the left chord $Z_i Z_l$ in anticlockwise direction.

Taking the left chord $Z_i Z_l$ as the side, we construct a rectangle $B_l$ with the length being $\rho$ and the width being $\rho/2$ inside the leaf area. In the same way, we use the right chord $Z_i Z_r$ to construct another rectangle $B_r$ inside the right area. As shown in Fig. 2 (a), the obtained two rectangle are symmetrical about the bisector of the angle $\alpha_{i,\rho}$. We use them to cut two rectangular patches from the leaf image $I$ and denote them as $P_l^{(1)}$ and $P_r^{(1)}$, respectively. Where we sample each of them to $256 \times 128$ pixels with subtracting the mean of the pixel values over the patch. By rotating the rectangles $B_l$ and $B_r$ by $\alpha_{i,\rho}/4$, in clockwise and counter-clockwise directions, respectively and using them to cut the leaf image $I$, we can achieve another two

rectangular patches as shown in Fig. 2 (b). We denote them as $P_l^{(2)}$ and $P_r^{(2)}$, respectively. Obviously, they are still symmetrical about the bisector of the angle $\alpha_{i,\rho}$. Continuing to rotate the rectangles $B_l$ and $B_r$ by $\alpha_{i,\rho}/4$, in clockwise and counter-clockwise directions, respectively and using them to cut the leaf image $I$, we can get two rectangular patches as shown in Fig. 2 (c), denoted by $P_l^{(3)}$ and $P_r^{(3)}$, respectively.

Collecting all the available three patch pairs, associated with the contour point $Z_i$, we can construct a composite structure denoted by

$$TPCS_{i,\rho} = \{(P_l^{(t)}, P_r^{(t)}), t = 1, 2, 3\}. \qquad (2)$$

We name it Triplet Patch-Pairs Composite Structure (TPCS). It has the following appealing characteristics which have potential to benefit the mining of discriminative deep features: (1) All the patch pairs in $TPCS_{i,\rho}$ have the common axis of symmetry, i.e., the bisectior of the angle $\alpha_{i,\rho}$ which makes them have tight spatial connections. (2) As shown in Fig. 3, TPCS varies with the change of the contour point $Z_i$ (resulting the variation of the angle $\alpha_{i,\rho}$) which make it have the merit of capturing the local structure information of leaf image. (3) The patches in TPCS are segmented from the leaf image in different orientations which enables the capturing of direction patterns. (4) The parameter $\rho$ of TPCS can be used to change the size of the patches in TPCS which potentially facilitating the extraction of multiscale deep extractions.

### 3.2 Local Deep Self-Similarity Descriptor

In this subsection, we use each patch pair in TPCS to measure the local self-similarity of leaf images in deep feature domain. For each pair of patches $P_l^{(t)}, P_r^{(t)}$ in TPCS, we feed them through a pre-trained CNN network to extract two feature maps, $f_l^t$ and $f_r^t \in R^{H \times W \times C}$, respectively, from the same convolutional layer, where $H \times W$ and $C$ are the spatial resolution and the number of the

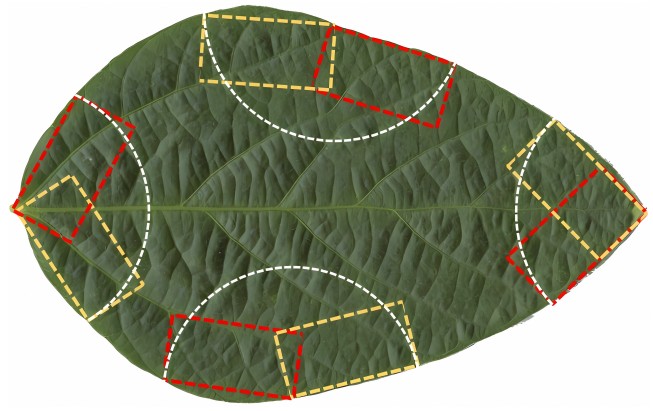

**Figure 3: An example to show that TPCS varies with the change of the contour point $Z_i$.**

channels of the feature map, respectively. According to the symmetric relationship between the patches $P_l^{(t)}, P_r^{(t)}$, the element $(u, v, k)$ of the feature map $f_l^t$ is symmetrical to the element $(u, W - v + 1, k)$ of the feature map $f_r^t$. By measuring the symmetric difference between the two feature maps, $f_l^t$ and $f_r^t$, we can get a new feature map $f_d^t \in R^{H \times W \times C}$ with each element calculated by

$$f_d^{(t)}(u, v, k) = (f_l^{(t)}(u, v, k) - f_r^{(t)}(u, W - v + 1, k))^2. \quad (3)$$

We conduct channel-wise average pooling against it to yield a feature vector $S^{(t)} \in R^C$. Fig. 4 presents a flow chart to illustrate the procedure of constructing the vector $S^{(t)}$. Then using the TPCS and a pre-trained CNN network, we get three feature vectors $S^{(t)}, t = 1, 2, 3$. Through normalizing them with L2-norm and concatenating them, we can build a descriptor

$$SD_{i,\rho} = \{S^{(t)}/\|S^{(t)}\|_2, t = 1, 2, 3\}, \quad (4)$$

associated with the contour point $z_i$ under the parameter $\rho$, to measure the local self-similarity of the leaf image. We tern it Local Deep Self-Similarity Descriptor.

### 3.3 SDePR Image-level Representations

**Multi-scale descriptors:** By changing the parameter $\rho$ of the local descriptor $SD_{i,\rho}$, we can extend it to multi-scale descriptors. Considering that the absolute scale parameter $\rho$ depends on the size of the leaf image, instead of directly using it, we introduce a relative scale parameter $\varphi \in [1, \dots, \Phi]$ which has the relation of $\rho = \Upsilon_i/2^\varphi$ with the parameter $\rho$ (see Equ. 1 for the definition of $\Upsilon_i$), where $\varphi$ is the index of the scale level and $\Phi$ is the number of the scale levels. For the contour point $Z_i$, let the relative scale parameter $\varphi$ vary from 1 to $\Phi$, we can generate $\Phi$ triplet patch-pairs composite structures: $\{TPCS_{i,1}, \dots, TPCS_{i,\Phi}\}$. As an example, Fig. 5 shows the first patch pairs in $TPCS_{i,1}$ and $TPCS_{i,2}$, respectively. We separately use $TPCS_{i,\Upsilon_i/2^\varphi}, \varphi = 1, \dots, \Phi$ to construct local deep self-similarity descriptors of $\Phi$ scale levels: $SD_i^{(\varphi)}, \varphi = 1, \dots, \Phi$.

**Local descriptors aggregation:** Since there are $N$ sample points in the leaf contour, for each scale $\varphi \in [1, \dots, \Phi]$, we can obtain $N$ local deep self-similarity descriptors $SD_i^{(\varphi)}, i = 1, \dots, N$, we simply

use the average pooling to aggregate them into a single feature vector

$$\overline{SD}^\varphi = \frac{1}{N} \sum_{i \in [0, N-1]} SD_i^{(\varphi)}. \quad (5)$$

By concatenating the aggregated descriptors of $\Phi$ scale levels $\overline{SD}^\varphi, \varphi = 1, \dots, \Phi$, we build an image-level representation which is a $3C \cdot \Phi-$ dimensional vector. As an image-level representation, it has the following characters: (1) According to the definition of the triplet Patch-Pairs composite structure (TPCS), the triple patch pairs and their spatial correlations does not depend on the rotation and translation of the leaf image. Therefore, the resulted descriptors $\overline{SD}^\varphi$ are invariant to the rotation and translation of leaf images. Since we use a relative scale parameter $\varphi$ instead of the absolute parameter $\rho$, the resulted descriptors $\overline{SD}^\varphi$ are also invariant to scaling of leaf images. (2) The TPCS is a symmetrical geometric structure and each patch pair is used to measure the local self-similarity of leaf images. Therefore, the resulted descriptors $\overline{SD}^\varphi$ can perfectly capture the structural information and spatial-contextual information. (3) The proposed image-level representation uses muti-scale features which make it have the ability to depict the leaf image from coarse to fine. (4) The proposed image-level representation is constructed from the activation of pre-trained CNN model and is training-free, i.e., does not depend on any training datasets which make it have strong generalization ability and discriminative power. Considering the unique characters of the proposed image-level representation, we name it Structural Deep Patch Representation (SDePR). Since SDePR is a single vector, we can measure the similarity between two leaf images by simply calculating their L1-distance for efficient leaf image retrieval.

### 3.4 Muti-Layer SDePRs Fusion

The ensemble of multiple layers of CNN models can boost the final recognition performance [31]. However, the feature vectors extracted from different layers generally have different dimensionalities. The simple concatenation of them may not make them complementary with each other to contribute to the performance improvement. Considering that developing a novel feature fusion scheme is not the focus of this study, we adopt a recent feature fusion scheme, KNN-HDFF, proposed by [5]. KNN-HDFF is a simple way originally designed for fusing handcrafted features and deep features. It considers the neighbouring information of different feature spaces in the normalization of distance measures. In this study, we use it to fuse the SDePRs constructed from different layers. Limited by the length of the article, we omit the introduction of its details (refer to [5] for the details of KNN-HDFF).

## 4 EXPERIMENTS

In this section, to gauge the performance of the proposed method for fine-grained leaf image retrieval (FGLIR), we conduct extensive experiments on the publicly available benchmarks and compare our method with the state-of-the-art methods with standard evaluation metrics.

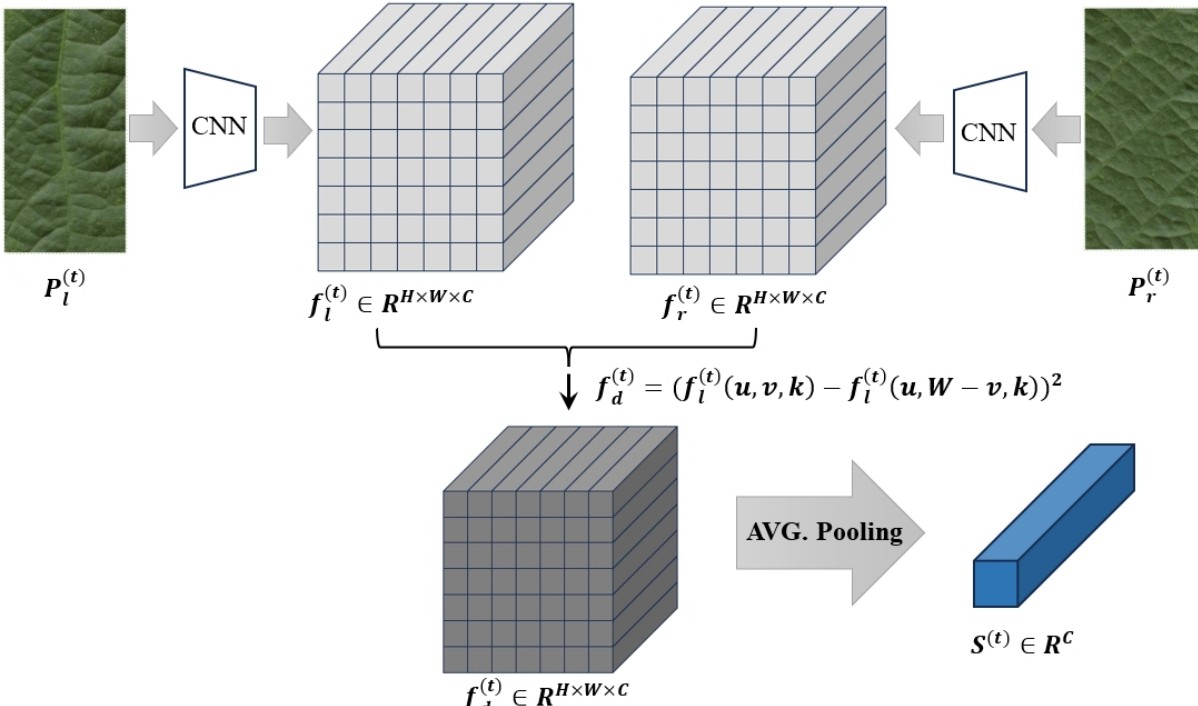

**Figure 4: A flow chart to illustrate the procedure of constructing the local deep self-similarity descriptor.**

## 4.1 Evaluation Metrics and Baselines

Two standard evaluation metrics, Bulls-eye test [5, 6, 10, 16, 27] and precision-recall (PR) curves [5, 6, 26] are used to quantify the retrieval performance of all the competing algorithms. Eight state-of-the-art image descriptors, Local RsCoM [29], SBT [5], FB-BDP [6], DSFH [17], ReSW [20], HeW-ResNet50 [19], DOLG [34], and SENet

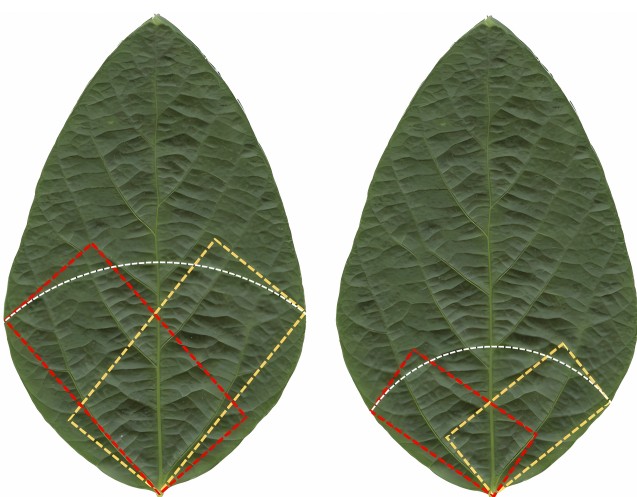

**Figure 5: An example to show the multicale TPCSs (only the first path pair is shown for simplified illustration).**

[14] are used as baselines. Among them, the first three descriptors are recently published handcrafted leaf image descriptors focusing on the characterization of fine-grained leaf images and report state-of-the-art performance on fine-grained leaf image recognition. While the last five descriptors are based on deep CNN features and achieve state-of-the-art performance on image retrieval tasks. DSFH [17] used the off-the-shelf features extracted from the fully-connected layer of pr-trained VGG-16 model. ReSW [20] used the the features of convolutional layers from the pre-trained or fine-tuned VGG-16 models. HeW-ResNet50 [19] extracted patch-level faetures from the last convolutional layer of the pre-trained and fine-tuned ResNet50 models. While as for DOLG [34] and SENet [14], they both took ResNet50 as backbone and consider the output of the convolutional layers for constructing their own network. Their networks require an additional dataset such as Google landmarks dataset V2 (GLDV2) [33] to train them. Their parameters all follow their original settings.

## 4.2 Implementation Details

All the competing methods are performed on a computer equipped with an intel i5-10500 CPU and a NVIDIA RTX 3070 GPU. Since our method focuses on developing training-free deep feature representation for FGLIR, a new challenging unsupervised pattern recognition task, we directly use the pre-trained CNN model and treat it as an generic feature extractor. VGG-16 is a popular pre-trained model which has been widely used for various pattern recognition tasks due to its great generalization ability and low computation cost. We therefore choose it as the extractor of feature maps for constructing

our SDePR image-level representation. We use the publicly available VGG-16 model (imagenet-matconvnet-vgg-verydeep-16) using the open-source library MatConvNet. The existing VGG-16 based methods generally use layers in the 5th convolutional blocks such as 'pool5', 'conv5-2' and 'conv5-3' for feature extraction. Different from them, besides the the use of 'pool5', we also consider the use of 'pool4' due to its preservation of richer spatial information over the deeper layers which benefit the capturing of structural and spatial-contextual information. We use the KNN-HDFF [5] to fuse the SDePRs from the 'pool4' and 'pool5' layers of pre-trained VGG-16 for image retrieval. The parameter settings for our method are: the number of the points sampled from the contour is $N = 64$ and the number of scale levels are $\Phi = 4$.

### 4.3 Soybean FGLIR

Soybeans are commonly known as one of the most important economic crops in the world. SoyCultivar200 [30] is a public available soybean leaf image dataset. It has 200 cultivars with each having 30 leaves collected from different parts of soybean plants: 10 samples from the upper part, 10 samples from the middle part, and 10 samples from the lower part. All the samples from the same part of soybean plants are grouped into a subset which makes the dataset divided into three subsets, Soy-Up, Soy-Mid, and Soy-Low, respectively. Each subset accordingly consists of 200×10=2000 single leaf patterns. Besides the above three subsets focusing on the testing of single leaf image patterns, this benchmark also designs another evaluation protocol on the use of joint leaf image patterns. In this protocol, all the 6000 leaves in the dataset are divided into 2000 groups with each containing three leaves of the same cultivar separately from the upper part, middle part and lower part of different soybean plants. Each group is treated as a joint leaf pattern and all of them form a set, Soy-Joint, consisting of 200×10=2000 joint leaf patterns. For a evaluated method, its feature descriptors extracted from the joint leaf patterns are concatenated as a single vector for image retrieval. Fig. 6 shows parts of leaf images of different cultivars from the three subsets, Soy-Up, Soy-Mid, and Soy-Low. More details about the SoyCultivar200 dataset refer to [30].

The Bull-eye scores of all the competing methods are summarized in Table 1. It can be seen that on the use of single leaf image patterns (performing on the test cases, Soy-Up, Soy-Mid and Soy-Low, respectively), the proposed methods achieves the scores of 65.59%, 64.28%, and 59.91% which are separately 4.55%, 6.09% and 3.77% higher than the other competing methods. While using joint leaf image patterns (Soy-joint), the proposed method achieves an exciting score of 92.64% which outperforms the other competing methods by 1.98%. We also report the results of our method on using 'pool4' or 'pool5' layer for soybean FGLIR. Although their scores are both lower than those of using the fusion of 'pool4' and 'pool5' layers, they are still significantly higher than those of all the other competing methods. We plot the PR curves for all the competing methods on the four test cases in Fig. 7. It can be clearly observed that on all the test cases, the proposed method consistently achieves the best PR curves.

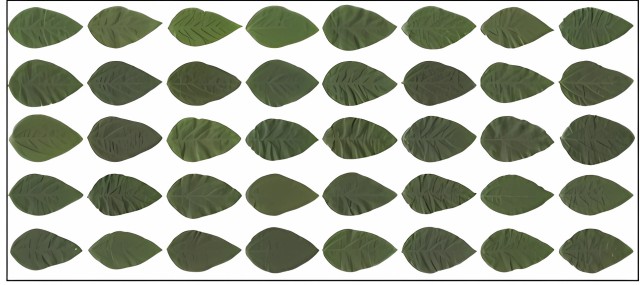

**(a) Soy-Up**

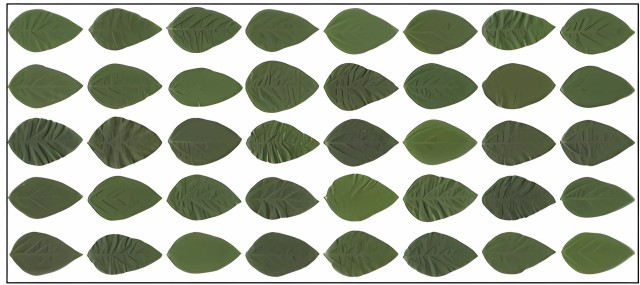

**(b) Soy-Mid**

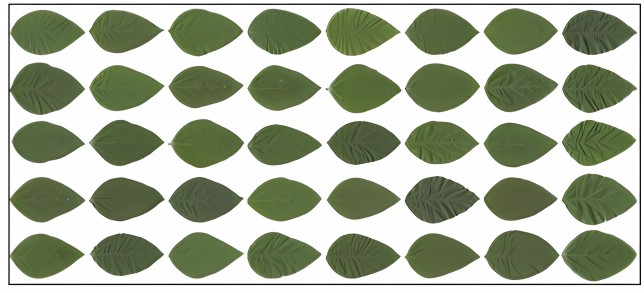

**(c) Soy-Low**

**Figure 6: Parts of Soybean leaf images of different cultivars from the three subsets of the SoyCul-tivar200 dataset.**

### 4.4 Peanut FGLIR

Peanut is another economic crop in the world which is widely grown in the tropics and subtropics, contributing to both small and large commercial producers. The PeanCultiar120 dataset [6] is another public available benchmark designed for FGLIR. It has 600 leaf images from 120 peanut cultivars with each consisting of 5 samples. Different from the SoyCultivar200, this dataset is designed only for testing the performance of single leaf image pattern. An example sample for each peanut cultivar is shown in Fig. 8.

The Bull-eye scores for all the competing methods are summarized in Table 2. As can be seen that the proposed method obtains the score of 59.93% which out-performers the other competing methods by 5.4%. We also report the scores of our method on using 'pool4' or 'pool5' layer for peanut FGLIR. Although their scores are both lower than that of using their fusion, they still achieve better scores over the all the other competing methods. In Fig. 9,

**Table 1: The Bull-eye scores (%) of all the competing methods on the three benchmarks, Soy-Up, Soy-Mid, Soy-Low of the SoyCultivar200 leaf image dataset [30].**

| Algorithm | Up | Mid | Low | AVERAGE |
|---|---|---|---|---|
| Local RsCoM | 42.77 | 43.11 | 41.60 | 70.44 |
| SBT | 47.57 | 48.40 | 47.92 | 81.69 |
| FB-BDP | 49.12 | 51.76 | 50.59 | 84.20 |
| DSFH | 41.52 | 41.82 | 40.35 | 69.90 |
| ReSW | 45.12 | 46.33 | 43.78 | 80.97 |
| HeW-Resnet50 | 48.07 | 49.13 | 45.23 | 80.42 |
| DOLG | 56.61 | 55.90 | **56.14** | 88.39 |
| SENet | 61.04 | 58.19 | 53.94 | 90.66 |
| **Proposed SDePR (Pool4)** | **61.55** | **60.63** | 57.10 | **91.76** |
| **Proposed SDePR (Pool5)** | **61.48** | **60.65** | 57.05 | **92.35** |
| **Proposed SDePR (Pool4+Pool5)** | **65.59** | **64.28** | **59.91** | **92.64** |

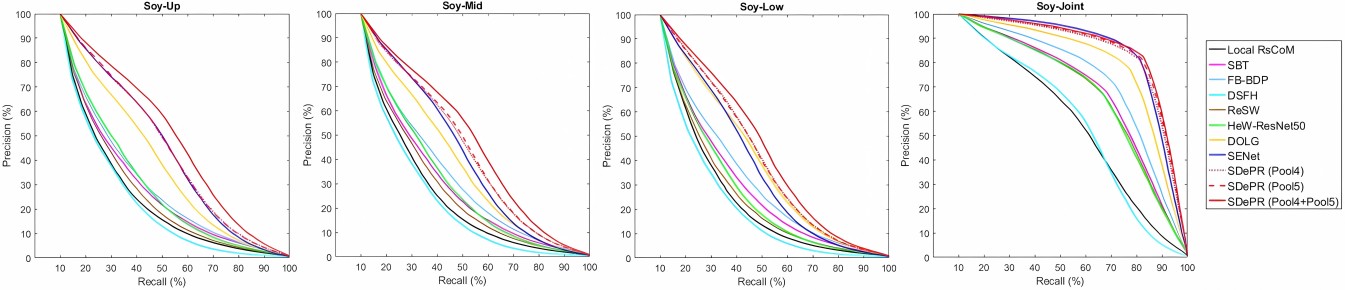

**Figure 7: The precision-recall curves for all the ten competing methods on the four test cases, Soy-Up, Soy-Mid, Soy-Low and Soy-Joint of the SoyCultivar200 leaf image dataset[30].**

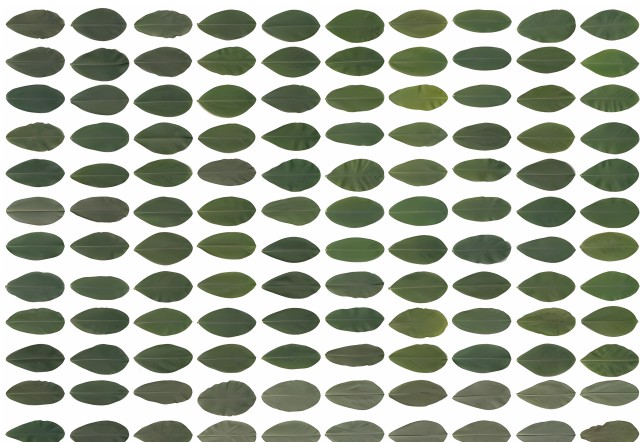

**Figure 8: 120 example peanut leaf images of the PeanCultivar120 dataset [6] (one example for each cultivar).**

we plot the PR curves of all the competing methods. It can be seen that the PR curve achieved by our method is obviously better than those of the other methods which consistently indicate its superior performance over the state-of-the-art methods on FGLIR.

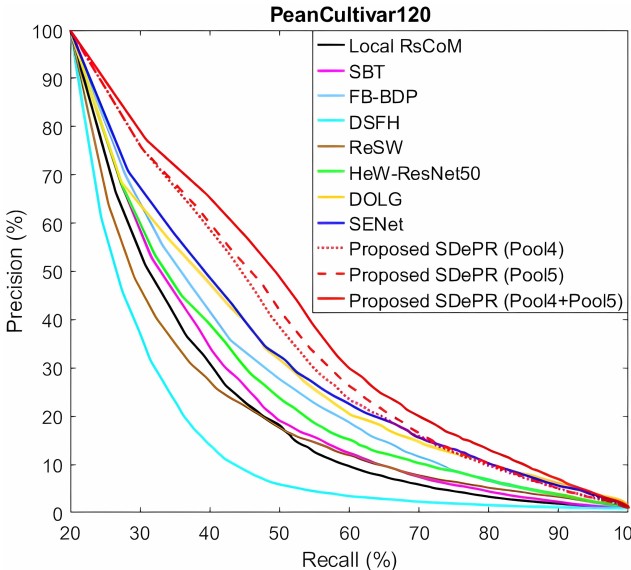

**Figure 9: The precision-recall curves of all the competing methods on the PeanCultivar120 leaf image dataset [6].).**

**Table 2: The Bull-eye scores (%) of all the competing methods on the PeanCultivar120 dataset.**

| Algorithm | Bull-eye scores (%) |
|---|---|
| Local RsCoM | 45.23 |
| SBT | 46.83 |
| FB-BDP | 51.80 |
| DSFH | 37.10 |
| ReSW | 44.73 |
| HeW-Resnet50 | 49.27 |
| DOLG | 53.77 |
| SENet | 54.53 |
| **Proposed SDePR (Pool4)** | **56.23** |
| **Proposed SDePR (Pool5)** | **57.93** |
| **Proposed SDePR (Pool4)** | **59.93** |

**Table 3: The retrieval scores of individually using the different pool lyaers.**

| layer | The size of feature map | Bull-eye scores (%) |
|---|---|---|
| Pool1 | $128 \times 64 \times 64$ | 45.27 |
| Pool2 | $64 \times 32 \times 128$ | 51.37 |
| Pool3 | $32 \times 16 \times 256$ | 55.43 |
| Pool4 | $16 \times 8 \times 512$ | 56.23 |
| Pool5 | $8 \times 4 \times 512$ | 57.93 |

## 4.5 Ablation Experiments

In this subsection, we use the PeanCultiar120 dataset to separately study the influences of the choice of layers in the pre-trained VGG-16 model. For the choice of layers, we first conduct a group of experiments on the individual use of 'poo1', 'poo2', 'poo3', 'poo4', and 'poo5'. The experimental results are summarized in Table 3. It can be seen that the features extracted from the deep layers work better than those of shallow layers due to the lack of high-level semantic information. However, due to the strong ability of our proposed SDePR on modelling the structural and spatial-contextual information, the use of middle layer ('pool3') for our method achieves the score 55.43% which still outperforms all the other competing methods. Next, we conduct another group of experiments to fuse the SDePR of 'pool5' with the SDePRs of 'pool4', 'pool3', 'pool2' and 'pool1'. The results are summarized in Table 4. It can be seen that compared with the individual use of 'pool5' layer, the fusions of 'pool5' with 'pool3' and 'pool4', respectively can both significantly improve the retrieval performance (resulting an increase of about 2%).

## 5 CONCLUSION

Fine-grained leaf image retrieval (FGLIR) is a new challenging unsupervised pattern recognition task. Due to the very high inter-class similarity across different cultivars, FGLIR tasks require powerful leaf image descriptors to handle the subtle difference among varieties within the same plant species. In this work, we make the first attempt to investigate the possible way to encode spatial structure and contextual information into the image-level representation for

**Table 4: The retrieval scores of fusing the SDePR of 'pool5' layer with the other 'pool' layers.**

| The fusion of 'Pool5' with others | Bull-eye scores (%) |
|---|---|
| Pool5 | 57.93 |
| Pool5+Pool4 | 59.93 |
| Pool5+Pool3 | 59.73 |
| Pool5+Pool2 | 57.00 |
| Pool5+Pool1 | 52.07 |

FGLIR. We design triplet patch-pairs composite Structure (TPCS) to extract patch-level deep features from the a pre-trained CNN model and measure the difference between the feature maps of the patch pair for constructing local deep self-similarity descriptors. The final aggregated local deep self-similarity descriptors, named Structural Deep Patch Representation (SDePR), have the strong ability to capture the spatial structure and contextual information of leaf images which make it more suitable for depicting fine-grained leaf images. We conduct extensive experiments to apply our proposed SDePR method to the public challenging fine-grained leaf image retrieval tasks which greatly improve the retrieval rates of the state-of-the-art methods.

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
