# OpenReview forum: "SDePR: Fine-Grained Leaf Image Retrieval with Structural Deep Patch Representation"
_acmmm.org/ACMMM/2024/Conference — MM2024 Poster_

### Official Review · Reviewer_3a32 · 2024-05-20

**Rating:** 5
**Confidence:** 2

**Summary:**

In this study, the authors for the first time investigate the possible way to mine the spatial structure and contextual information from the activation of the convolutional layers of CNN networks for FGLIR.

**Strengths:**

The paper for the first time investigate the possible way to mine the spatial structure and contextual information from the activation of the convolutional layers of CNN networks for FGLIR. For achieving this goal, authors design a novel geometrical structure, named Triplet Patch-Pairs Composite Structure (TPCS).

**Limitations:**

(1) There are some errors in the paper, such as “SDEPR” in the abstract, bold in Table 1, naming of the last algorithm in Table 2, and spelling errors in Table 3, which need to be carefully corrected.
(2) A leaf image is uniformly sampled with 64 points, together with multi-scale variations, yielding more local images. Therefore, how efficient is SDePR?

**Suitability:**

3

---

### Official Review · Reviewer_ZixM · 2024-05-21

**Rating:** 3
**Confidence:** 2

**Summary:**

This paper proposes a Structural Deep Patch Representation (SDePR) for fine-grained leaf image retrieval (FGLIR). The proposed method investigates the possible way to mine the spatial structure and contextual information from the activation of the convolutional layers of CNN networks for FGLIR. Extensive experimental results and visual analysis show that the proposed method outperforms the state-of-the-art handcrafted visual features and deep retrieval models.

**Strengths:**

1. The topic of this paper is interesting. A Triplet Patch-Pairs Composite Structure (TPCS) is introduced to extract the texture information of leaf images.
2. This paper utilizes the pre-trained CNN method to extract convolutional features from leaf image patches that represent spatial structure and contextual information.
3. Extensive experiments demonstrate the effectiveness of the proposed method in fine-grained leaf image retrieval tasks.

**Limitations:**

1. What are the differences between the proposed Triplet Patch-Pairs Composite Structure (TPCS) in this paper and the existing FGLIR methods?
2. The authors should present figures demonstrating the differences between the proposed method and existing FGLIR methods to highlight the innovations of the proposed method.
3. This paper lacks an overall flowchart demonstrating the proposed method to ensure that readers can understand the entire process of how leaf image retrieval is implemented.
4. In the ablation experiments, the authors should present comparative results showing whether or not TPCS and multi-scale features are used.
5. What are the advantages of the proposed method compared to inputting the entire leaf image into a CNN network for feature extraction?
6. The authors should provide source code to enhance the replicability of the proposed method.

**Suitability:**

2

---

### Official Review · Reviewer_b2yj · 2024-05-21

**Rating:** 2
**Confidence:** 4

**Summary:**

The paper presents a novel approach for fine-grained leaf image retrieval, focusing on the subtle differences among various cultivars within a species. It introduces a method called Structural Deep Patch Representation (SDePR), which utilizes convolutional neural network (CNN) activations to capture spatial and contextual information from leaf images. This approach leverages a Triplet Patch-Pairs Composite Structure (TPCS) to construct local deep self-similarity descriptors, allowing for effective retrieval across different scales and transformations. Extensive experiments on public datasets demonstrate that SDePR outperforms existing methods.

**Strengths:**

In general, the paper is well written and the motivation for this submission is easy to understand, but the novelty is limited.

**Limitations:**

1. For Figure 4, do these two cnn's share parameters? Or are they trained separately by p_l^(t) and  p_r^(t).
2. For Figure 5, the description above mentions performing a scale rotation of the ρ, and what effect does this rotation angle have on the experimental results?
3. For ablation experiments, the description above mentions performing a scale rotation of the ρ, and what effect does this rotation angle have on the experimental results?
4.  What do the results of random sampling look like? For example, random blocks of leaves were collected as p_l^(t) and r_r^(t)
5. The method involves constructing multi-scale descriptors and employing complex structures like the Triplet Patch-Pairs Composite Structure (TPCS). The computational cost and efficiency of these processes are not discussed.
6.  Although the paper claims superior performance over state-of-the-art methods, the comparison mainly focuses on other leaf image retrieval methods. There is a lack of comparison with the latest general image retrieval techniques that use more advanced deep learning strategies

**Suitability:**

2

---

### Official Review · Reviewer_Lt3z · 2024-05-22

**Rating:** 3
**Confidence:** 3

**Summary:**

This article conducts research on fine-grained leaf image retrieval using unsupervised learning. By designing a triple patch-pair compositive structure (TPCS) module, a large number of patches on the leaf surface are extracted. Then, local structure properties and spatial correlations are obtained through CNN, and image retrieval is further achieved through the fusion of multi-scale and multi-layer CNN features. The effectiveness of the proposed method has been demonstrated through a large number of comparative experiments.

**Strengths:**

(1) This proposed method is special. Unsupervised methods have better generalization compared to supervised ones.

(2) Comparative experiments on two datasets have demonstrated the effectiveness of the features proposed in this paper.

(3) This paper is well written and easy to understand.

**Limitations:**

(1) Pretrained CNN is crucial for extracting leaf features. This paper chooses imagenet-matconvnet-vgg-verydeep-16 as the feature extractor. Could you directly initialize the weights of CNN randomly as baseline, and choose other datasets to train VGG-16 from scratch, and then use it as the feature extractor for comparison. You could choose leaf/plant datasets with similar data sizes, as well as other datasets unrelated to leaves, such as Cats/Dogs/Cars. This comparison may reflect the importance of the pretrained CNN feature extractor.

(2) Although this article is a training-free method, TPCS extracts a large number of patches, resulting in a high time complexity for extracting CNN features for each leaf. could you include the comparison results of testing time consumption in Table 1 or Table 2.

(3) N and φ should be discussed in the ablation study.

(4) Minor errors, for example, are both Line 826 and Line 828 the results of "Proposed SDePR (Pool4)"? Spelling issues, such as "spe-ces" in line 86, "belong-ing" in line 87, "represneta- tion" in line 140, "da-tasets" in line 150, et al.

I am willing to revise my rating based on the authors’ feedback.

**Suitability:**

3

---

### Meta-Review · Area_Chair_dWrq · 2024-07-02

**Recommendation:** Accept (Poster)
**Confidence:** 5

**Metareview:**

The paper received mixed scores. The average score is on the positive side, however there still are concerns.